# FLEXIBLE AND ADAPTABLE SUMMARIZATION VIA EXPERTISE SEPARATION

## ABSTRACT

A proficient summarization model should exhibit both *flexibility* – the capacity to handle a range of in-domain summarization tasks, and *adaptability* – the competence to acquire new knowledge and adjust to unseen out-of-domain tasks. Unlike large language models (LLMs) that achieve this through parameter scaling, we propose a more parameter-efficient approach in this study. Our motivation rests on the principle that the general summarization ability to capture salient information can be shared across different tasks, while the domain-specific summarization abilities need to be distinct and tailored. Concretely, we propose MoeSumm, a Mixture-of-Expert Summarization architecture, which utilizes a main expert for gaining the general summarization capability and deputy experts that selectively collaborate to meet specific summarization task requirements. We further propose a max-margin loss to stimulate the separation of these abilities. Our model's distinct separation of general and domain-specific summarization abilities grants it with notable flexibility and adaptability, all while maintaining parameter efficiency. MoeSumm achieves *flexibility* by managing summarization across multiple domains with a single model, utilizing a shared main expert and selected deputy experts. It exhibits *adaptability* by tailoring deputy experts to cater to out-of-domain few-shot and zero-shot scenarios. Experimental results on 11 datasets show the superiority of our model compared with recent baselines and LLMs. We also provide statistical and visual evidence of the distinct separation of the two abilities in MoeSumm[1].

## 1 INTRODUCTION

Text generation has made impressive progress in recent years (Ouyang et al., 2022). The task of abstractive summarization, aiming to produce a concise, fluent, and faithful summary, has become a research hotspot due to its broad application prospect (Wang et al., 2022; Chen et al., 2023). Herein, we outline two key capabilities that an intelligent summarization system should possess. The first is *flexibility*, indicating the system's competence to be readily applied to a variety of in-domain tasks. A flexible summarization system should be proficient in summarizing various types of content, such as news articles and scientific papers. The second ability is *adaptability*, for acquiring new knowledge and adapting to unseen out-of-domain summarization tasks. For example, a medical summarization model trained prior to 2019 needs to be able to adapt and acquire knowledge about COVID-19.

Existing pretrained summarization models typically use a *one-model-for-one-domain* approach, training separate models on individual datasets each optimized for a specific domain (See et al., 2017; Li & Liang, 2021). However, this strategy hampers their flexibility as a model tailored for one domain may underperform in others (Fu et al., 2023). Alternatively, recent LLMs like GPT-3 (Brown et al., 2020) and GPT-3.5 exhibit remarkable summarization performance, powered by vast data volumes and computational resources (Yang et al., 2023). However, these *one-large-model-for-all-domains* frameworks have their drawbacks, including being closed-source, costly, and susceptible to data leakage (Tian et al., 2023). Furthermore, their inability to edit or scale the knowledge embedded within them once trained results in limited adaptability to fresh knowledge (Cheng et al., 2023).

Different from previous works, this paper aims to improve summarization flexibility and adaptation in a parameter-efficient way, resulting in a *one-small-model-for-all-domains* approach. Our motivation

---

[1]Our code is attached and will be released with the camera-ready version.

stems from the need for a general summarization capability to distill key input information and a specialized adaptability to refine this information in line with specific summarization requirements such as language style, summary length, and conciseness. By sharing the general ability, a summarization model avoids redundant learning for each domain and focuses on common features, while separating specialized abilities ensures tailored, high-quality summaries without unnecessary complexity. Correspondingly, we propose a Mixture-of-Expert Summarization (*MoeSumm*) model, where a main expert captures salient information, and deputy experts work with the main expert to adapt the extracted summary information to the different domains. Specifically, we choose to incorporate expert separation by adapting the feed-forward neural networks (FFNs) in pretrained models. The main expert collaborates with selected deputy experts to form an FFN. These deputy experts, chosen by a dataset-aware gating function, are designed to learn dataset-aware summarization abilities. To prevent the model from over-relying on the main expert and collapsing into a single model, we propose a max-margin loss, where the margin is defined as the prediction difference brought by the deputy experts. Due to its decouple attribute, MoeSumm can naturally adapt to out-of-domain few-shot domains, where only the deputy experts need to be fine-tuned. MoeSumm can also be used in zero-shot settings, where we can utilize the main expert to give a general summary.

We validate the effectiveness of MoeSumm in 3 settings (in-domain, out-of-domain few-shot, and zero-shot) across 11 benchmark datasets. The datasets are from various domains (news, academic papers, social media posts, etc.), varying in input and output lengths, levels of abstractiveness, and language style. Experiment results show that our MoeSumm outperforms all baseline models in most of the metrics. In addition, we demonstrate the separation of general and different specialization abilities through comprehensive experiments, which also provide an explanation for the generation.

Overall, our main contributions are: (1) We propose MoeSumm, a parameter-efficient summarization model that is applicable to a variety of in-domain summarization tasks and is well-suited for out-of-domain few-shot and zero-shot summarization scenarios. The model's parameter efficiency is ensured by the shared general summarization ability. (2) To achieve the above goal, we design MoeSumm with a mixture-of-expert structure and a max-margin loss to distinguish the roles of the different experts. (3) Experimental results demonstrate that MoeSumm brings substantial improvements over strong baselines in both in-domain and out-of-domain scenarios across benchmark datasets.

## 2    RELATED WORK

**Summarization.** Most existing summarization models adopt a *one-model-for-one-domain* approach. For example, Yu et al. (2021); Chen et al. (2022) finetune all the parameters in the model for each target dataset. Fonseca et al. (2022) first conduct content selection and then output the summary. While these models achieve good performance on specific summarization tasks, they tend to struggle in maintaining consistent effectiveness across a variety of domains (Wang et al., 2019) and in sustaining a general ability (Fu et al., 2023). Another line of research aims to propose *one-model-for-all-domains*. Wang et al. (2019) developed a multi-domain extractive summarization model and examined the impact of domain discrepancies on extractive summarization performance. Despite demonstrating flexibility, the model didn't effectively address the adaptability challenge associated with out-of-domain tasks. Most recently, large language models such as GPT-3 have shown impressive flexible summarization abilities, made possible by vast data and computational resources. In contrast, we aim for a parameter-efficient approach that addresses both flexibility and adaptability.

**Mixture-of-Experts Models.** MoE models were initially proposed to increase the model's capacity while maintaining a constant computational cost during inference, where a fixed number of experts are adaptively activated by an input during training and inference (Zuo et al., 2021; Mustafa et al., 2022). Typically, a trainable gate in MoE determines the activation of experts, often resulting in an imbalance with most inputs routed to a single expert. Various restriction losses have been proposed to address this (Lewis et al., 2021; Fedus et al., 2021). Here, we introduce a dataset-aware selector to tackle the imbalance issue. Gao et al. (2022) proposed a matrix product operator to reconstruct the matrix in the expert layer and increase model capacity. In contrast, our work achieves enhancing parameter efficiency by sharing the general summarization ability. In the domain of summarization, Ravaut et al. (2022) proposed one of the few works that related to MoE, where they used an MoE architecture to rerank summary candidates, which is different from our work.

## 3 BACKGROUND

We base our summarization model on the prevalent Transformer architecture (Vaswani et al., 2017), comprised of an encoder and decoder, each with repeated Transformer blocks. Each block has a multi-head self-attention sub-layer and a two-layer feed-forward neural network (FFN). Suppose the self-attention output is $\mathbf{A}$. Then, the FFN outputs $\mathbf{X}$ by:

$$\mathbf{H} = \sigma\left(\mathbf{A}\mathbf{W}_1 + \mathbf{b}_1\right), \quad \mathbf{X} = \mathbf{H}\mathbf{W}_2 + \mathbf{b}_2, \tag{1}$$

where $\mathbf{W}_1 \in \mathbb{R}^{d \times d_h}, \mathbf{W}_2 \in \mathbb{R}^{d_h \times d}, \mathbf{b}_1 \in \mathbb{R}^{d_h}$ and $\mathbf{b}_2 \in \mathbb{R}^d$ are weights of the FFN, $\sigma$ is the activation function, $d$ is the embedding dimension, and $d_h$ is the hidden dimension of the FFN.

Mixture-of-Experts was firstly proposed to facilitate conditional computation and increase the parameter count without altering the floating point operations for each input (Shazeer et al., 2017). Essentially, MoE models consist of multiple expert layers similar to the Transformer layers. Each of these layers contains a self-attention mechanism and multiple FFNs (Eq. 1) in parallel, namely "experts", denoted as $\{E_i\}_{i=1}^N$. Each expert has its own set of learnable weights. To keep the computational cost constant, a gating network $G$ outputs a sparse $N$-dimensional vector to route each token via a few experts.

Similar to Eq. 1, we denote the output of the attention mechanism as $\mathbf{A}$. For each $\mathbf{a}_s$ (the $s$-th row of $\mathbf{A}$) that corresponds to the $s$-th input token, the corresponding output $\mathbf{x}_s$ of FFNs is:

$$\mathbf{x}_s = \sum_{i \in \mathcal{T}} G_i\left(\mathbf{a}_s\right) E_i\left(\mathbf{a}_s\right). \tag{2}$$

Here, $\mathcal{T} \subset \{1 \cdots N\}$ is the activated set of experts that have the largest $G_i$ values, and $G_i\left(\mathbf{a}_s\right)$ denotes the probability of selecting expert $E_i$.

Various approaches have been proposed to compute $G_i$ and construct $\mathcal{T}$. A classic method, proposed by Shazeer et al. (2017), calculates $G_i$ by a weighted matrix $\mathbf{W}$ based on the input $\mathbf{a}_s$:

$$G_i\left(\mathbf{a}_s\right) = \left[\text{softmax}\left(\mathbf{a}_s\mathbf{W}\right)\right]_i, \tag{3}$$

where $\mathbf{W} \in \mathbb{R}^{d \times N}$. This method, however, has two major drawbacks: (1) It often leads to a load imbalance problem, where $\mathbf{W}$ collapses, causing nearly all the inputs to be routed to the same expert (Fedus et al., 2021; Zuo et al., 2022). (2) The gating function lacks awareness of the input dataset's diversity, an important source of information that reflects the attributes of the inputs.

## 4 THE PROPOSED MOESUMM MODEL

In this section, we first present an algorithm that adapts an MoE into our MoeSumm model. Then, we detail how MoeSumm can be used in out-of-domain few-shot and zero-shot scenarios.

The overall framework of our model is shown in Fig. 1(a). Our model includes a main expert used for all datasets and a dataset-aware expert selector to choose suitable deputy experts. This dataset-aware selection method overcomes the previously mentioned limitations by ensuring that cases with similar attributes are routed to the proper deputy experts based on the dataset information.

Let $N^p$ denote the number of deputy experts, and $\mathbf{a}_{s,e}$ be the token representation in the $s$-th position of the input sequence from dataset $e$ after the attention process. Let's consider trainable weight matrices $\mathbf{W}_e \in \mathbb{R}^{d \times N^p}$ corresponding to each dataset $e$. We multiply the input $\mathbf{a}_{s,e}$ with the dataset-specific weight matrix $\mathbf{W}_e$ to incorporate data information in the gating mechanism, yielding the routing logits:

$$c_e\left(\mathbf{a}_{s,e}\right) = \mathbf{a}_{s,e}\mathbf{W}_e, \tag{4}$$

where $c_e(\mathbf{a}_{s,e}) \in \mathbb{R}^{N^p}$. To obtain the routing probabilities, we normalize the routing logits using a softmax over the $N^p$ deputy experts. The gate value for the $i$-th deputy expert is then given as:

$$G_{i,e}\left(\mathbf{a}_{s,e}\right) = \text{softmax}[c_e\left(\mathbf{a}_{s,e}\right)]_i. \tag{5}$$

We can now select the top-$k$ gate values for routing the token. Following previous works (Gupta et al., 2022; Zuo et al., 2022), we constrain the gating method to route each token to only the top-1 expert FFN:

$$p = \text{argmax}_i G_{i,e}(\mathbf{a}_{s,e}), \quad g_p = G_{p,e}(\mathbf{a}_{s,e}),$$

MoeSumm Training with Expertise Separation

(a) on multiple high-resource datasets:

Main Expert

Deputy Expert 1

Dataset-aware Expert Selector

Deputy Expert 2

Deputy Expert 3

docs from different datasets

summaries

(b) on low-resource out-of-domain datasets:

Main Expert

Deputy Expert 1

Dataset-aware Expert Selector

Deputy Expert 2

Deputy Expert 3

Fixed

Figure 1: Training MoeSumm under different settings. (a) Training the expert selector and all experts on multiple high-resource datasets. (b) Fine-tuning only the expert selector and the deputy experts on low-resource datasets.

where $g_p$ is the highest score. Following Eq. 1, we integrate the outputs from main and deputy experts, guided by the gate score:

$$\mathbf{H} = \sigma\left(\left[\mathbf{A}\mathbf{W}_1^m + \mathbf{b}_1^m; g_p\left(\mathbf{A}\mathbf{W}_1^p + \mathbf{b}_1^p\right)\right]\right), \tag{6}$$

$$\mathbf{X} = [\mathbf{W}_2^m; \mathbf{W}_2^p]\mathbf{H} + \mathbf{b}_2^m, \tag{7}$$

where $[;]$ denotes the concatenation operation, and superscript $m$ and $p$ denote parameters from the main and selected deputy expert, respectively.

In the above formulation, the dataset-aware gating function $\mathbf{W}_e$ learns to route input tokens to specialized experts. Importantly, the experts don't have a direct relationship with the datasets, but depend on the input context, encouraging information sharing among all experts and datasets.

**Max-margin Loss.** The intrinsic difference between our MoeSumm and standard MoE is the roles assigned to experts. MoeSumm features a main expert that acquires a generalized summarization skill adaptable to diverse datasets, and deputy experts that specialize in handling cases with specific attributes. Given the difficulty of defining general and specialized summarization targets, we propose a max-margin loss. This strategy aims to prevent the model from over-relying on the main expert, thereby ensuring the contributions of deputy experts aren't overshadowed.

As illustrated in Fig. 2, we first define the margin as the difference between the predicted probabilities of the full model (with main and deputy experts) and the main model (using only main expert):

$$m_t = P_t^{\text{full}}\left(y_t\right) - P_t^{\text{main}}\left(y_t\right), \tag{8}$$

where $y_t$ is the $t$-th token in the summary, and $P_t^{\text{full}}$ and $P_t^{\text{main}}$ denote the predicted probability of the $t$-th token by the full model and the main model, respectively. Intuitively, a large $m_t$ suggests that the full model significantly outperforms the main model, highlighting the valuable contributions of deputy experts and the effective collaboration between main and deputy experts. If $m_t$ is small, there are two possibilities. One is that both the full and the main models perform well, resulting in similar predicted probabilities (both $P_t^{\text{full}}$ and $P_t^{\text{main}}$ are high). The other possibility is that the main expert is not good enough but overconfident, thus, leading to subpar performance of both the full and main models (both $P_t^{\text{full}}$ and $P_t^{\text{main}}$ are low).

Hence, we present the max-margin loss $\mathcal{L}_m$, which adds a coefficient to the margin:

$$\mathcal{L}_m = \sum_{t=1}^{n_y}\left(1 - P_t^{\text{full}}\right)\left(1 - m_t^5\right)/2, \tag{9}$$

where we abbreviate $P_t^{\text{full}}(y_t)$ as $P_t^{\text{full}}$. The term $(1 - m_t^5)/2$ is a monotonically decreasing non-linear function with respect to $m_t$, which ensures that the minimization of $\mathcal{L}_m$ maximizes $m_t$. We choose a Quintic function (fifth power) here as it offers more stability (Miao et al., 2021). The first factor $(1 - P_t^{\text{full}})$ accounts for the two scenarios illustrated in Fig. 2. When $P_t^{\text{full}}$ is high, the summarization model performs well, needing minimal optimization on $m_t$. This is reflected by $(1 - P_t^{\text{full}})$, which acts as a small coefficient of $m_t$. On the other hand, when $P_t^{\text{full}}$ is low, a large coefficient $(1 - P_t^{\text{full}})$ encourages the maximization of $m_t$, so that the correct target word can be predicted with the help of deputy experts. The overall loss function of MoeSumm is a combination of text generation loss and max-margin loss.

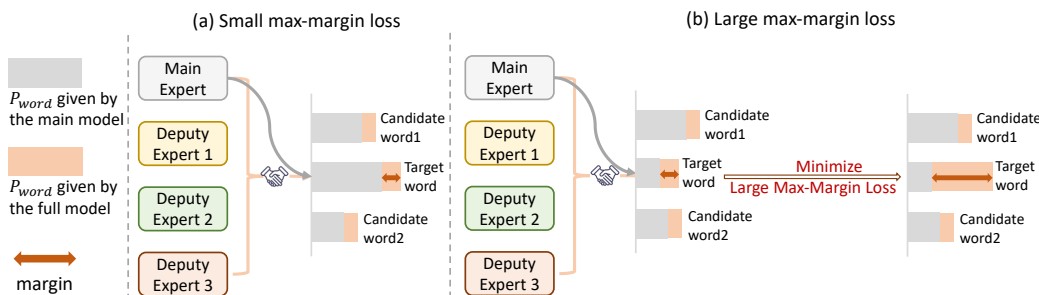

Figure 2: Examples illustrating the max-margin loss $\mathcal{L}_m$ in two scenarios. (a) $\mathcal{L}_m$ is small when the main expert performs well, where both $P_{\text{word}}^{\text{full}}$ and $P_{\text{word}}^{\text{main}}$ for the target word surpassing other candidates. (b) $\mathcal{L}_m$ is large when the main model cannot perform well. In this scenario, minimizing the max-margin loss can maximize the margin $m_t$, thus preventing the overconfidence of the main model and stimulating deputy experts to learn to predict the correct target word.

**Adaptability of MoeSumm.** Due to its inherent separation of general and specialization ability, MoeSumm has the adaptability to handle few-shot and zero-shot summarization scenarios for out-of-domain data. Firstly, we can reuse the main expert and only fine-tune the deputy experts along with the expert selector to quickly adapt MoeSumm to a low-resource dataset, as shown in Fig. 1(b). Moreover, in a zero-shot scenario where no training data is available, we can rely solely on the main expert to generate summaries. This approach is possible as the main expert is competent at producing general summaries, especially when the model lacks prior knowledge about the target domain.

## 5 EXPERIMENTS

**Dataset and Evaluation setting.** In the standard evaluation setting, MoeSumm is trained on a *Mix dataset*[2] comprising of three widely-used summarization datasets: CNN/DM (Hermann et al., 2015), WikiHow (Koupaee & Wang, 2018), and PubMed (Cohan et al., 2018), selected for their diverse domains and varying source and target lengths. For out-of-domain few-shot evaluation, MoeSumm is fine-tuned as shown in Fig. 1(b). We use a small number of samples from XSum (Narayan et al., 2018), AESLC (Zhang & Tetreault, 2019), and Reddit (Kim et al., 2019) for fine-tuning the expert selector and deputy experts, and assess the fine-tuned MoeSumm on the corresponding testing samples. For zero-shot evaluation, there is no fine-tuning, the full-scale MoeSumm trained on Mix dataset is tested on unseen datasets including Gigaword (Napoles et al., 2012), BillSum (Kornilova & Eidelman, 2019), arXiv (Cohan et al., 2018), BIGPATENT (Sharma et al., 2019), and MultiNews (Fabbri et al., 2019). We provide a dataset attribute table in Appendix.

**Comparison Methods.** BART (Lewis et al., 2020) is a well-known pretraining sequence-to-sequence model. We assess BART's performance in two different settings. BART-mix: training a single model on the entire Mix dataset, and BART-indiv: training separate models on individual datasets within the Mix dataset. For BART-indiv, we utilize its CNN/DM version for zero-shot evaluation and fine-tune it in the few-shot scenario. We also employ a naive flat MoE baseline, where there is no main expert, and expert selection has no dataset information. The obtained models after mixed training are directly used for the zero-shot test. We also show the superiority of our model compared with a prompt-tuning approach Prefix (Li & Liang, 2021), an adapter-based baseline (Huh & Ko, 2022), and GPT-3.5. Details can be found in Appendix.

**Implementation Details.** We implemented our experiments in Huggingface on 4 NVIDIA A100 GPUs. We used the BART-large as the pretrained language model by default. The expert dimension $d_h$ is 512 and the deputy expert number is 3 by default, for balancing between model complexity and performance. The impact of expert dimension in MoeSumm is analyzed in detail in Appendix. We used Adam optimizer with $\epsilon$ as 1e-8 and $\beta$ as (0.9, 0.999). The learning rate is set to 3e-5. The warm-up is set to 500 steps for all experiments. The batch size is set to 8 with gradient accumulation

---

[2]Further experiments with varied mix datasets and architectures are in Discussion and Appendix section.

Table 1: Performance in in-domain and out-of-domain scenarios. **Bold** numbers indicate statistically *significant* improvements over the FlatMoe, using a two-tailed paired t-test (Dror et al., 2018) at a significance level of 0.05. (+%) is the average percentage improvement in ROUGE over FlatMoe.

| Dataset | BART-mix $R_1$ / $R_2$ / $R_L$ / BS | BART-indiv $R_1$ / $R_2$ / $R_L$ / BS | FlatMoe $R_1$ / $R_2$ / $R_L$ / BS | MoeSumm $R_1$ / $R_2$ / $R_L$ / BS / (+%) |
|---|---|---|---|---|
| *Trained on Mix dataset, in-domain test*: | | | | |
| CNN/DM | 43.84/20.78/40.53/89.01 | 44.16/21.28/40.90/88.16 | 44.09/21.03/40.95/89.02 | **45.01/21.75/41.91/89.26** (+2%) |
| PubMed | 44.52/17.96/39.62/86.26 | 44.57/17.96/39.70/86.25 | 44.74/17.86/39.86/86.38 | **45.40/18.44/40.54/86.60** (+2%) |
| WikiHow | 46.49/20.44/44.90/90.68 | 46.96/20.93/45.41/90.81 | 46.83/20.34/45.05/90.71 | 46.75/**21.35/45.33/90.89** (+2%) |
| *Trained on Mix dataset, out-of-domain zero-shot test*: | | | | |
| Gigaword | 26.21/8.94/23.11/86.04 | 25.14/8.52/22.42/85.45 | 25.80/9.06/22.65/85.95 | **26.57/9.32/23.89/85.87** (+4%) |
| BillSum | 43.16/18.78/36.36/84.22 | 41.19/18.04/34.47/83.98 | 42.84/18.95/35.74/84.12 | **43.65/19.22/36.45/84.16** (+2%) |
| arXiv | 41.41/14.18/36.53/85.39 | 39.58/13.14/33.70/84.02 | 42.05/14.46/37.26/84.66 | **43.91/15.51/38.55/85.60** (+5%) |
| BIGPATENT | 34.82/10.17/29.15/83.78 | 32.55/8.95/27.59/83.82 | 35.05/10.37/29.13/83.66 | **37.02/11.10/31.01/84.18** (+5%) |
| MultiNews | 28.69/9.44/25.65/85.25 | 27.86/9.34/25.17/83.71 | 28.97/9.73/26.12/85.02 | **31.62/10.49/28.37/85.48** (+8%) |
| *Fine-tuned for out-of-domain few-shot test*: | | | | |
| XSum10 | 32.21/9.01/23.74/88.76 | 31.81/8.82/23.33/88.68 | 32.65/9.06/23.87/88.76 | **33.15/10.22/24.42/89.21** (+5%) |
| XSum100 | 35.17/12.05/27.52/89.74 | 34.69/11.77/27.36/89.67 | 35.25/12.03/27.87/89.89 | **35.58/13.06/28.06/89.94** (+3%) |
| AESLC10 | 26.17/12.72/23.39/84.38 | 26.56/12.81/23.81/84.18 | 26.47/12.83/23.76/84.48 | **27.48/14.32/25.53/86.04** (+7%) |
| AESLC100 | 30.44/17.20/28.64/84.98 | 30.01/15.26/26.98/84.94 | 31.12/17.34/29.39/86.03 | **32.87/17.96/30.92/86.54** (+5%) |
| Reddit10 | 19.39/6.47/17.44/86.97 | 17.56/5.58/15.74/85.45 | 20.03/6.89/18.89/87.22 | **21.74/8.00/20.75/88.09** (+11%) |
| Reddit100 | 21.62/8.37/20.45/87.94 | 19.44/7.19/17.34/86.00 | 23.31/9.93/22.59/88.13 | **25.57/11.45/24.34/88.67** (+10%) |

steps of 4. When decoding a summary, we used beam search with a beam size of 4, and the vocabulary size of the model is 50,625. See Appendix for more details.

**Evaluation Metrics.** We employ standard ROUGE F1 (Lin, 2004): ROUGE-1 ($R_1$), ROUGE-2 ($R_2$), and ROUGE-L ($R_L$), each indicating the matches of unigram, bigrams, and the longest common subsequence, respectively. We also use BERTScore (BS) (Zhang et al., 2020) to calculate semantic similarities between the summaries. Beyond these automatic evaluation metrics, we assess system performance by human judgments.

## 5.1 MAIN RESULTS OF PERFORMANCE COMPARISON

**Performance on In-domain Test.** The first three rows in Tab. 1 show the performance of baseline models and our model on in-domain test. We first observe that BART-indiv performs better than BART-mix in most metrics. This is expected as the three datasets have distinct attributes that can confuse a mixed single model. Secondly, FlatMoe performs comparably to BART-indiv and outperforms BART-mix due to its expert structure. Finally, our MoeSumm model leverages the three datasets more effectively, achieving significantly better results across four metrics on all three datasets. Specifically, it outperforms BART-mix by 3%/5%/3% RG-1/RG-2/RG-L scores on CNN/DM respectively. This shows that our MoeSumm model, with its hierarchical mixture-of-expert structure, can effectively utilize combined training datasets from different domains as a method of data augmentation.

**Performance in Out-of-domain Zero-shot Scenarios.** The adaptability of models on unseen tasks is reported in the second block in Tab. 1. Models are tested on Gigaword, BillSum, arXiv, BIGPATENT, and MultiNews datasets, which encompass various fields such as news, academic papers, patents, and bills. BART-mix outperforms BART-indiv, highlighting the benefits of multi-dataset learning for adaptability. FlatMoe does not show significant improvement compared with BART-mix, indicating that flat MoE structure cannot improve the generalization ability of the model. MoeSumm demonstrates significantly superior adaptability, outperforming baselines in all metrics. It is worth noting that, in the zero-shot scenario, MoeSumm introduces no extra parameters compared to the basic BART model, as the expert selector and deputy experts are not used.

**Performance in Out-of-domain Few-shot Scenarios.** In Tab. 1, the third section shows results from scenarios where only 10/100 samples from datasets such as XSum, AESLC, and Reddit (spanning news, email, and post domains) are available for fine-tuning. These results are averaged over 10 independent runs. BART-mix significantly outperforms BART-indiv, similar to the zero-shot setting.

Table 3: Ablation study of MoeSumm when Dataset Information (DI) in expert selector and max-margin loss ($\mathcal{L}_m$) are removed. **Bold** numbers indicate significant improvements over the second-best. (+%/+%) is the average percentage improvement in ROUGE over w/o DI and w/o $\mathcal{L}_m$.

| Test Dataset | Test $R_1$ / $R_2$ / $R_L$ / BS | MoeSumm w/o DI $R_1$ / $R_2$ / $R_L$ / BS | MoeSumm w/o $\mathcal{L}_m$ $R_1$ / $R_2$ / $R_L$ / BS | MoeSumm $R_1$ / $R_2$ / $R_L$ / BS / (+%) |
|---|---|---|---|---|
| CNN/DM | in-domain | 44.42/21.13/40.67/88.57 | 43.91/20.91/40.69/88.83 | **45.01/21.75/41.91/89.26** (+2%/+3%) |
| BIGPATENT | 0-shot | 36.60/10.62/30.85/83.93 | 36.38/10.25/30.04/83.59 | **37.02/11.10/31.01/84.18** (+2%/+4%) |
| AESLC | 100-shot | 32.72/17.16/30.09/85.04 | 32.46/17.33/30.86/86.10 | **32.87/17.96/30.92/86.54** (+3%/+2%) |

MoeSumm achieves better performance than the strong baseline FlatMoe, demonstrating the effectiveness of our hierarchical expert structure in distinguishing general and specialized summarization abilities across various low-resource scenarios.

**Human Evaluation.** We also conducted a human evaluation of our model to balance the potential bias of automated metrics in assessing summarization quality (Schluter, 2017). We randomly sampled 50 test instances from the CNN/DM, Gigaword, and XSum datasets. Following the methodology proposed by Liu et al. (2022), but with a threefold larger evaluation scale, we presented three Ph.D. evaluators with an article and its corresponding system-generated summaries. They were asked to rate these summaries based on Succinctness, Informativeness, and Fluency. The score ranges from one to three, where three is the best. The averaged results

Table 2: Human evaluation results of three models, in terms of succinctness, informativeness, and fluency of generated summaries.

| Model | Succ | Inform | Flu |
|---|---|---|---|
| BART-mix | 2.37 | 2.34 | 2.07 |
| FlatMoe | 2.42 | 2.39 | 2.28 |
| MoeSumm | **2.56** | **2.62** | **2.44** |
| GPT-3.5 | 2.33 | 2.65 | 2.61 |

are shown in Tab. 2. Our model outperforms the baseline models BART-mix and FlatMoe in all metrics. The kappa statistics are 0.41, 0.44, and 0.45 for fluency and consistency respectively, and indicate moderate agreement between annotators. A t-test between our model and FlatMoe confirmed the statistical significance of these results. Further case studies can be found in the Appendix.

**Comparison with GPT-3.5.** Our human evaluation also consists of a comparison with GPT-3.5. As shown in Tab. 2, MoeSumm displays superior succinctness and comparable informativeness to GPT-3.5, while GPT-3.5 gives more fluent text. Examples in the Appendix, such as Fig. 14, show that MoeSumm produces more concise summaries, whereas GPT-3.5 outputs more conjunctions such as 'while' and 'although'. Furthermore, GPT-3.5 often produces inferred sentences that enhance comprehensibility at the expense of brevity. This aligns with previous findings (Yang et al., 2023) that ChatGPT generally opts for more extended summaries. Moreover, we provide ROUGE comparison in Fig. 11 in Appendix, which reveals MoeSumm's advantage in both in-domain and out-of-domain tasks. Taking into account that GPT-3.5 boasts 300 times more parameters, coupled with the recent insights (Liu et al., 2023; Zhang et al., 2023) regarding the alignment of the ROUGE metric with human annotations, MoeSumm's performance is commendable.

## 5.2 DISCUSSION

**Ablation Study.** We removed the dataset information in the expert selector and max-margin loss to evaluate their impact on MoeSumm during in-domain test, out-of-domain few-shot and zero-shot test. When the dataset information was removed, the deputy experts were selected only based on the input content. As shown in Tab. 3, this leads to a notable performance drop in all test scenarios, underscoring the importance of introducing our dataset-aware selection. Additionally, eliminating the max-margin loss resulted in a 4% ROUGE-2 score reduction in zero-shot and few-shot settings, indicating its role in distinguishing the functions of main and deputy experts.

**Analysis on Expertise Specialization.**

**1) Different deputy expert exhibits unique characteristics.** We first study this problem from qualitative aspect. Take deputy expert (DE) #1 and #3 for example, we found that DE #3 excels in generating scholarly summaries, while DE #1 adeptly describes news events. DE #3 is inclined to generate longer and more complex sentences while DE #1 usually generates simpler sentences. Below are two *randomly* selected examples generated by our model using different deputy experts on MultiNews and PubMed datasets:

| Dataset | MoeSumm with DE#1 | MoeSumm with DE#3 |
|---|---|---|
| MultiNews | Scott Stevens was fired from his job after his employer discovered he was embezzling money from his company to fund his gambling habit. | He gave his wife instructions to avoid responsibility for his losses and keep her credit intact: she was to deposit a check for $4,000; move her funds into a new checking account; decline to pay the money he owed the Bellagio casino in Las Vegas; disregard his credit-card debt; file her tax returns; sign up for Social Security survivor benefits; and have him cremated. |
| PubMed | while no study has examined the influence of anxiety on cognition in patients living with pd by directly comparing groups of pd patients with and without anxiety [author annotation: with no detailed information on the experiments.] | using a cross-sectional design, we compared 17 pd participants with anxiety and thirty-three participants without anxiety on the mini-mental state exam (mmse), the parkinsonism rating scale (prs), and the revised barthel index (rbans). |

Consequently, we conduct a quantitative analysis. MoeSumm with DE #1 tends to generate shorter sentences (15 words on average), and MoeSumm with DE #3 can generate longer sentences (37 words on average). We also find that with DE #1, the model obtains a performance of 43.34/16.03/38.29 RG-1/RG-2/RG-L, whereas with DE #3, the ROUGE performance is improved by 1.4/1.61/1.57 on PubMed. These observations correspond to Fig. 3 in our paper, where DE #1 is more frequently chosen for CNN/DM, and DE #3 is predominantly selected for PubMed.

**2) Deputy experts are utilized differently.** Second, we assessed how deputy experts are specialized for each dataset. As illustrated in Fig. 3, MoeSumm avoids the pitfall of expert collapse, a situation where inputs are channeled to a single expert Zuo et al. (2022); Roller et al. (2021). The utilization distribution can also provide an intuitive understanding of the domain-specific abilities each expert acquires. For example, DE #1 is proficient in handling news and thus more selected by CNN/DM dataset, DE #2 is good at summarizing user-generated content and thus largely used by WikiHow dataset, and DE #3 is adept in medical information for handling PubMed dataset.

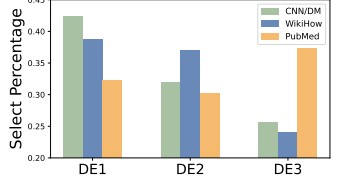

Figure 3: Distribution of selected Deputy Experts (DE) associated with three different datasets.

**3) Deputy expert utilization reflects the dataset attributes.** Finally, we statistically assessed specialized abilities by examining dataset attributes and expert utilization distributions. If the deputy experts have indeed learned specialized abilities, datasets with similar attributes should select similar deputy experts. Representing each dataset with a vector [*coverage, density, compression, domain*], where Grusky et al. (2018) defines the first three and *domain* denotes "news", "scholar", or "user content", we mapped each to its deputy expert uti-

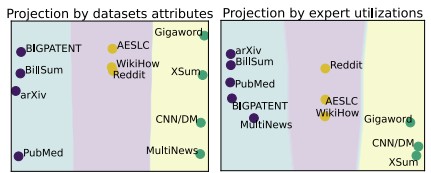

Figure 4: Projection comparison between dataset attributes and their deputy expert utilization distribution.

lization post MoeSumm fine-tuning on each dataset. Fig. 4 showcases PCA projection and clustering by attributes and expert utilization. It is clear that similar datasets are projected in near space, demonstrating that the MoeSumm learns specialized abilities via the deputy experts. Notably, MultiNews aligns closely with other datasets with long documents, indicating that deputy experts are sensitive to document length.

**Analysis on General-Specific Expertise Separation.** We next investigated whether the general and specialized abilities are indeed separated in MoeSumm. First, we compare MoeSumm and MoeSumm w/o any deputy experts (solely the main expert). The expectation is that MoeSumm w/o any deputy experts lacks the ability to adapt to target summary length and language style. Fig. 5(a), depicts how the generated summary length varies with the maximum decoding restrictions on PubMed dataset. It is evident that MoeSumm w/o deputy experts lacks information regarding the target length, while MoeSumm efficiently halts the generation process to produce an optimized summary length. This finding highlights that the deputy experts store domain-specific knowledge.

Fig. 5(b) presents a *randomly sampled* case where the full MoeSumm model and MoeSumm w/o deputy experts summarize a PubMed paper on Parkinson's disease patients, abbreviated as "pd". The results show that MoeSumm w/o deputy experts struggles to comprehend and accurately employ the key phrase, whereas MoeSumm properly mentions that the experiment was conducted on 17 Parkinson's disease patients with anxiety and 34 patients without anxiety. This indicates that the deputy experts carry the specialized ability specific ability to understand domain-specific terms.

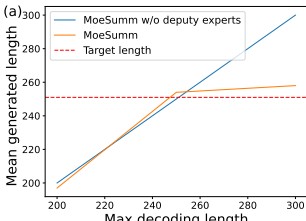 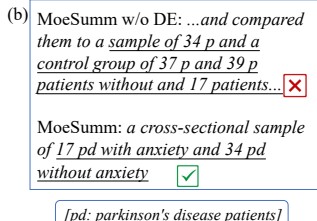 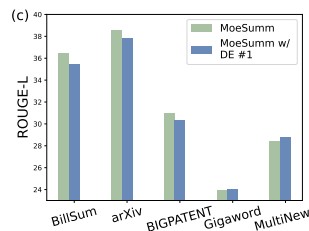

Figure 5: Analysis of the separation of general and deputy abilities. (a) Comparing MoeSumm and MoeSumm w/o any deputy experts (solely the main expert) on the length of the generated summary. (b) Performance of MoeSumm and MoeSumm w/o deputy experts (DE) for domain-specific context. (c) The performance of MoeSumm and MoeSumm with only DE #1 on five datasets.

Finally, Fig. 5(c) shows the performance of MoeSumm with the main expert and only DE #1 across five datasets. According to the analysis from Fig. 3, DE #1 is proficient at handling news articles. This is reflected in Fig. 5(c). Compared to MoeSumm with all experts, the model equipped with only DE #1 excels in summarizing news domain datasets like Gigaword and MultiNews, but underperforms in other domains like scholarly papers and bills. This comparison confirms the general summarization proficiency of the main expert, and the flexibility of MoeSumm in selecting suitable deputy experts to complement the main expert, resulting in effective performance across diverse datasets and domains.

**Impact of Training Scale.** Our main experiment sets the Mix dataset scale to 3. In the meantime, we were interested to see how the generalization and specialization abilities of MoeSumm change with more training datasets. Concretely, we sequentially added MultiNews, AESLC, and Reddit to the Mix dataset, expanding it to 3 to 6 sub-datasets. Fig. 6 presents the ROUGE-L performance of BART-mix and MoeSumm on in-domain WikiHow and out-of-domain arXiv tests

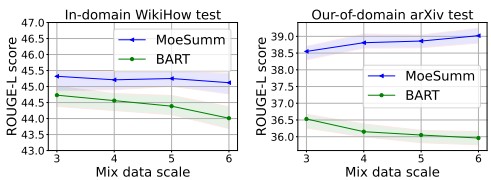

Figure 6: Influence of training data scale on in-domain and out-of-domain dataset.

with these varied scales. BART-mix's performance drops on the in-domain and out-of-domain datasets as Mix dataset size grows, highlighting its struggle with varied domain training due to its non-specific design. Conversely, MoeSumm consistently excels, retaining performance on WikiHow and bettering on arXiv, showcasing its generalization improves with training dataset variety. Although MoeSumm's potential could scale with more datasets, it's bound by model size. Hence, we look forward to evaluating the potential of our framework with a larger backbone.

**Robustness & Efficiency.** Previous research indicated that prompt-based fine-tuning could lead to high variance (Köksal et al., 2022). We test MoeSumm and Prefix using 10 different seeds, resulting in varied training data choices. Results in Fig. 7 show that MoeSumm significantly enhances robustness compared to Prefix. For *parameter scale*, in a standard setting, our model introduces additional 24M parameters for each input, which is notably fewer than the 62M added by Prefix. This count is also substantially less than using individual BART models for each dataset, which would require 387M parameters. For *time efficiency*, MoeSumm takes only about 10% longer than FlatMoe.

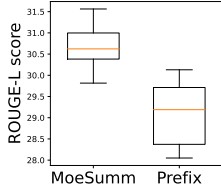

Figure 7: Robustness.

## 6 CONCLUSION

In this paper, we enhanced the flexibility and adaptability of summarization by introducing a parameter-efficient model based on a modified mixture-of-experts structure. The model consists of a main expert that learns general ability to identify important information, and deputy experts that adapt to domain-specific summary styles. Our model can be readily applied to diverse summarization datasets and adapted for out-of-domain situations. Experimental results showed that our model outperforms strong baselines. In the future, we would like to test the performance of our architecture on larger pretrained language models.

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

## A  APPENDIX

### A.1  LIMITATIONS

We discuss the limitations of our framework as follows:

(1) In this work, due to limited computation resources, we mainly investigate the performance of the unified model trained by three datasets, and have a rough evaluation of the unified model trained by up to six datasets. However, in real-world applications, an ideal unified model should be trained on more datasets from various domains. This requires a larger model with more parameters. Hence, we look forward to proposing an enhanced summarization model trained on more datasets in the future.

(2) The pre-training and fine-tuning summarization data used in this study are publicly available. Despite the fact that the original datasets were filtered during construction, some contents, such as news accounts of violent crimes and events, may contain uncomfortable descriptions.

### A.2  IMPLEMENTATION DETAILS

We implement our experiments in Huggingface (Wolf et al., 2019) on 4 NVIDIA A100 GPUs. We used the BART-large (Lewis et al., 2020) as the pretrained language model. The expert dimension is 512 and the deputy expert number is 3 by default. We use Adam optimizer with $\epsilon$ as 1e-8 and $\beta$ as (0.9, 0.999). The learning rate is set to 3e-5. The warm-up is set to 500 steps for all experiments. The batch size is set to 8 with gradient accumulation steps of 4. When decoding a summary, we used beam search with a beam size of 4, and the vocabulary size of the model is 50,625. The encoded

length is 1024, and the maximum decode length is 300. We select CNN/DM, PubMed, and WikiHow as primary settings to train the unified summarization model, since they cover different text domains and length attributes.

**Baseline implementations.** To our best knowledge, no prior works use mixture of expert structure for abstractive summarization. Hence, we design and implement FlatMoe ourselves, where there is no main expert, and expert selection has no dataset information. For baseline Prefix, we use the code provided by the authors (Li & Liang, 2021)[3]. The performance of Prefix on XSum is slightly different from the originally reported result (Li & Liang, 2021). Similar observations have been found here[4]. Nonetheless, Prefix is still a reasonably evaluated baseline. We undertook our own implementation of Light, as the original work (Huh & Ko, 2022) did not provide the code. Notably, our version of Light outperforms the reported results in certain metrics, reaffirming the credibility of our reimplementation for evaluation comparisons. For the significance t-test, we use the code provided by Dror et al. (2018)[5].

**Training details.** For BART-indiv baseline, it is separately trained and evaluated by three datasets. Its CNN/DM-version is used in the zero-shot evaluation, and fine-tuned in the few-shot scenario. We choose CNN/DM-version since it achieves better performance.

For the other models in full-scale training, we use the Mix dataset to train the unified models. The obtained models after mixed training are directly used for the zero-shot test. For few-shot testing, only the prefix in Prefix and expert-related parameters are fine-tuned.

## A.3   DATASET ATTRIBUTES

Table 4: Statistics of the datasets used in the experiments. DL and SL denotes document length and summary length.

| Datasets | Domain | DL | SL | Size |
|----------|--------|-----|-----|------|
| CNN/DM | News | 810 | 56 | 287,226 |
| XSum | News | 431 | 23 | 204,045 |
| MultiNews | News | 2,103 | 263 | 44,972 |
| Gigaword | News | 30 | 8 | 3,803,957 |
| arXiv | Scholar | 4,938 | 220 | 196,807 |
| PubMed | Scholar | 3,016 | 203 | 174,134 |
| BIGPATENT | Scholar | 3,540 | 110 | 1,341,362 |
| BillSum | Scholar | 1,285 | 177 | 18,949 |
| Wikihow | User-generated context | 579 | 61 | 230,843 |
| Reddit | User-generated context | 342 | 9 | 122,933 |
| AESLC | User-generated context | 233 | 27 | 14,436 |

## A.4   PERFORMANCE ON OTHER ARCHITECTURES

Apart from BART-large, we also test our MoeSumm structure on BART-base and PEGASUS-large architecture in Tab. 5 and Tab. 6. The results show the generalization ability of our framework that does not depend on specific model scales or architectures. We choose the BART-large structure in our main experiment due to its better performance.

---

[3]https://github.com/XiangLi1999/PrefixTuning
[4]https://github.com/XiangLi1999/PrefixTuning/issues/2
[5]https://github.com/rtmdrr/testSignificanceNLP

Table 5: Performance comparison of BART-base and our MoeSumm based on BART.

| Dataset | Setting | BART-base | MoeSumm (BART-base) |
|---------|---------|-----------|---------------------|
| CNN/DM | in-domain | 42.45/19.52/39.23/88.52 | **43.68/20.30/40.37/88.75** (+4%) |
| arXiv | 0-shot | 39.56/12.64/34.84/84.86 | **41.99/13.77/37.05/85.19** (+5%) |
| AESLC | 100-shot | 27.74/15.71/26.66/85.88 | **29.71/17.14/28.54/86.46** (+5%) |

Table 6: Performance comparison of PEGASUS and our MoeSumm based on PEGASUS.

| Dataset | Setting | PEGASUS | MoeSumm (PEGASUS-version) |
|---------|---------|---------|---------------------------|
| CNN/DM | in-domain | 43.24/20.26/40.08/88.98 | **44.32/21.13/41.58/89.20** (+2%) |
| arXiv | 0-shot | 39.48/13.47/35.54/84.92 | **41.95/14.48/37.51/85.16** (+4%) |
| AESLC | 100-shot | 29.17/16.97/27.39/84.36 | **30.99/18.10/29.19/85.71** (+5%) |

## A.5 COMPARISON WITH MORE BASELINES

**Prefix** (Li & Liang, 2021) is a prompt-tuning approach that keeps BART frozen and optimizes a sequence of continuous task-specific vectors appended to the original tokens, denoted as prefix.

**Light** (Huh & Ko, 2022) is a lightweight meta-learning adapter inserted into the attention mechanism of BART, which is designed for low-resource scenarios. The performance is displayed in Tab. 11. Our model surpasses this recent robust baselines, primarily attributed to the expertise separation design we've incorporated.

## A.6 IMBALANCE PROBLEM IN MOE

we conduct new experiments to see if other imbalance-related techniques can be used to solve the imbalance problem in our scenario. Specifically, we choose the embedding method proposed by Pham et al. (2023), which introduces task embedding, additional task gate, and MoE gate to selector the adaptors. The results show that the baseline and our methods achieve similar balanced distributions but with fewer parameters.

Table 7: Ablation study on imbalance problem.

| Model | Deputy Expert #1 | Deputy Expert #2 | Deputy Expert #3 |
|-------|------------------|------------------|------------------|
| MoeSumm | 42% | 32% | 26% |
| MoeSumm (adapter-version) | 40% | 35% | 24% |

## A.7 IMPACT OF TRAINING SCALE

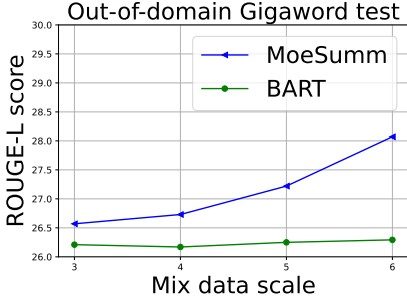

Figure 8: Influence of training data scale on out-of-domain Gigaword dataset.

## A.8 Impact of Expert Dimension

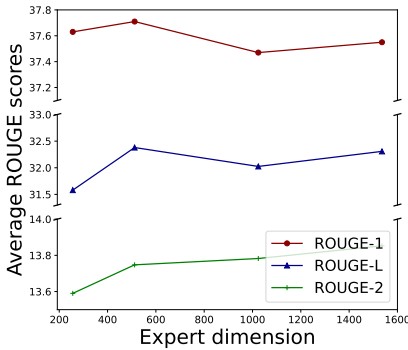

Figure 9: Ablation study on out-of-domain datasets in terms of expert dimension.

We next examine the effect of deputy expert setting on the generalization ability of unified summarization model. We choose to perform experiments in out-of-domain setting, since it can reflect the generalization ability of summarization models. The average of ROUGE-1, ROUGE-2, and ROUGE-L scores on four out-of-domain datasets, including Gigaword, BillSum, arXiv, and BIGPATENT are illustrated in Fig. 9. When the dimension increases, the capacity for deputy experts increases, and correspondingly, less is learned by the main expert. Our model exhibits peak performance with a dimension of 512. Moreover, MoeSumm continues to demonstrate solid performance under various parameter settings, reflecting the model's robustness.

## A.9 Deputy Expert Utilization

As illustrated in Fig. 10, MoeSumm avoids the pitfall of expert collapse, a situation where inputs are channeled to a single expert - an issue observed in related works by Zuo et al. (2022); Roller et al. (2021). The utilization distribution can also provide an intuitive understanding of the domain-specific abilities each expert acquires. For example, deputy expert #1 is proficient in handling news and thus more selected by CNN/DM dataset, deputy expert #2 is good at summarizing user-generated content and thus largely used by WikiHow dataset, and deputy expert #3 is adept in medical information for handling PubMed dataset.

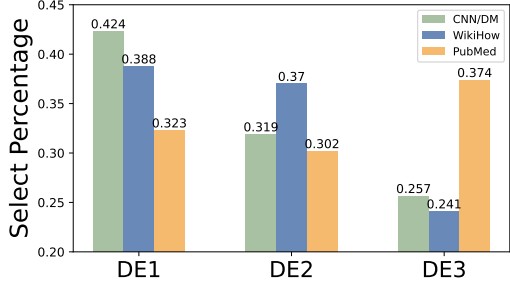

Figure 10: Selected percentages of different Deputy Experts (DE).

## A.10 Adaptability and Efficiency of MoeSumm

Let's denote the number of experts as $N^p$, the number of layers as $L$, and the number of parameters in each FFN expert as $P_f$. Consequently, the total quantity of expert parameters within the model can be calculated as $L \times N^p \times P_f$. It's important to note that these experts are shared across all datasets; hence, augmenting the number of datasets does not affect the count of expert parameters.

Conversely, the gating network is cognizant of the dataset and its parameter count increases with the inclusion of more training datasets. If we define $H$ as the dimension of the hidden state and $T$ as the number of datasets, then the quantity of gating parameters can be expressed as $L \times N^p \times H \times T$.

In practical scenarios, the hidden state dimension and the number of datasets are typically far less than the number of FFN parameters, i.e., $H \times T \ll Pf$. Hence, the augmentation of training datasets results in a comparatively smaller increase in parameters, especially when contrasted with the parameters inherent in standard feed-forward Transformer networks.

In Table 8 we show the actual parameter number as the training data scale grows. It can be seen that further scaling up the training data does not significantly increase the number of parameters.

| Model Name | Parameter Number |
|---|---|
| BART-large | 387.46M |
| Prefix | 449.42M (+16%) |
| MoeSumm (3 datasets) | 411.77M (+6%) |
| MoeSumm (4 datasets) | 411.84M (+6%) |
| MoeSumm (5 datasets) | 411.91M (+6%) |
| MoeSumm (6 datasets) | 411.98M (+6%) |

Table 8: Comparison of Parameter Numbers. The percentage indicates the parameter improvement compared to BART-large.

## A.11 COMPARISON AND EVALUATION WITH LLM

For using GPT-3.5 for summarization, the prompt we use is in the following format:

*Article: [article]*
*Summarize the above article in N sentences.*

Table 9: ROUGE performance of GPT-3.5 and our model in full-scale, zero-shot, and low-resource training scenarios.

| Dataset | GPT-3.5 $R_1$ / $R_2$ / $R_L$ | MoeSumm $R_1$ / $R_2$ / $R_L$ |
|---|---|---|
| CNN/DM | 39.98/14.67/35.66 | 46.59/22.64/43.23 |
| PubMed | 40.42/15.12/36.75 | 45.38/19.19/40.79 |
| XSum | 21.20/6.64/17.42 | 23.77/5.39/18.47 |
| AESLC | 35.83/15.24/29.52 | 31.03/16.51/30.60 |
| Gigaword | 25.08/8.29/21.61 | 26.63/9.72/23.93 |
| BillSum | 46.87/23.00/34.60 | 43.46/18.78/36.05 |
| arXiv | 40.23/12.04/35.96 | 43.86/15.41/38.56 |
| BIGPATENT | 37.87/10.95/31.91 | 35.85/11.06/29.85 |
| MultiNews | 36.18/10.63/31.97 | 32.26/10.41/28.72 |

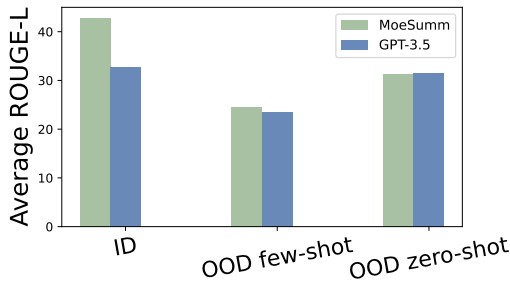

Figure 11: Performance of our model and GPT-3.5 in in-domain (ID) and out-of-domain (OOD) testing scenarios in ROUGE scores.

In §5.1, we report the human evaluation result. Here in Fig. 11 Tab. 9, and Fig. 11, it can be seen that MoeSumm outperforms GPT-3.5 in in -domain datasets and achieves comparable ROUGE scores to GPT-3.5 in out-of-domain scenarios. Taking into account that GPT-3.5 boasts 300 times more parameters, coupled with the recent insights by Liu et al. (2023); Zhang et al. (2023) regarding the alignment of the ROUGE metric with human annotations, MoeSumm's performance is notably commendable.

We also conduct an evaluation with the latest GPT-4. The instruction is:

*score summary1, summary2, and summary3 given the article in respect to fluency, with one (\*) to three stars (\*\*\*), where one star means bad and three stars means good. only return the stars. summary1: {summary1} summary2: {summary2} summary3: {summary3} article: {article}*

Table 10: Evaluation scores by GPT-4.

| Model | Succ | Inform | Flu |
|---|---|---|---|
| BART-mix | 2.32 | 2.36 | 2.68 |
| MoeSumm | 2.63 | 2.66 | 2.89 |
| GPT-3.5 | 2.81 | 2.90 | 2.95 |

It can be seen that GPT-4 adopts a more lenient standard in evaluating summaries, resulting in higher scores compared to our human evaluation. Meanwhile, the ranking it provides aligns with our human assessment, affirming the reliability and effectiveness of both evaluation methods and highlighting our model's advantage over the baseline.

Table 11: Performance of baselines and our model in low-resource training scenarios. Numbers in **bold** mean that the improvements to the second-best performance are statistically significant for $\alpha = 0.05$. (+%/+%) is the average percentage improvement in ROUGE over Prefix and Light.

| Dataset | Prefix $R_1$ / $R_2$ / $R_L$ / BS | Light $R_1$ / $R_2$ / $R_L$ / BS | MoeSumm $R_1$ / $R_2$ / $R_L$ / BS |
|---|---|---|---|
| XSum10 | 32.50/9.84/23.95/88.93 | 32.29/10.14/24.24/89.03 | **33.15/10.22/24.42/89.21** (+3%/+2%) |
| XSum100 | 35.20/12.74/27.57/89.73 | 35.39/12.90/27.83/89.80 | **35.58/13.06/28.06/89.94** (+2%/+1%) |
| AESLC10 | 26.45/13.08/24.26/84.84 | 26.59/13.42/24.53/85.06 | **27.48/14.32/25.53/86.04** (+6%/+5%) |
| AESLC100 | 31.58/16.83/29.11/85.87 | 32.02/17.64/29.64/85.85 | **32.87/17.96/30.92/86.54** (+6%/+3%) |
| Reddit10 | 19.95/6.88/17.89/87.03 | 20.24/7.24/18.60/87.47 | **21.74/8.00/20.75/88.09** (+14%/+10%) |
| Reddit100 | 23.64/9.89/22.53/87.62 | 24.47/10.34/23.06/88.03 | **25.57/11.45/24.34/88.67** (+11%/+7%) |

## A.12  CASE STUDY

See the selected case from CNN/DM, Gigaword, and XSum datasets in Fig. 12-14.

---

**Document:** A SkyWest Airlines flight made an emergency landing in Buffalo, New York, on Wednesday after a passenger lost consciousness, officials said. The passenger received medical attention before being released, according to Marissa Snow, spokeswoman for SkyWest. **She said the airliner expects to accommodate the 75 passengers on another aircraft to their original destination – Hartford, Connecticut** – later Wednesday afternoon. The Federal Aviation Administration initially reported a pressurization problem and said it would investigate. **Snow said** there was no indication of any pressurization issues, and the FAA later issued a statement that did not reference a pressurization problem. SkyWest also said there was no problem with the plane's door, which some media initially reported. Flight 5622 was originally scheduled to fly from Chicago to Hartford. The plane descended 28,000 feet in three minutes. "It would feel like a roller coaster – when you're coming over the top and you're going down," CNN aviation analyst Mary Schiavo said, describing how such a descent would feel. "You know that these pilots knew they were in a very grave and very serious situation."

**Ground Truth Summary:** FAA backtracks on saying crew reported a pressurization problem. One passenger lost consciousness. The plane descended 28,000 feet in three minutes.

**Baseline BART-mix:** SkyWest Airlines flight makes emergency landing in Buffalo, New York. Passenger received medical attention before being released, SkyWest spokeswoman says. **75 passengers on another aircraft expected to fly back to Hartford, Connecticut.**

**Baseline Prefix:** NEW: The passenger received medical attention before being released, a SkyWest spokeswoman says. NEW: There was no indication of any pressurization issues, **she says**. Flight 5622 was originally scheduled to fly from Chicago to Hartford. The plane descended 28,000 feet in three minutes.

**GPT-3.5:** A SkyWest Airlines flight made an emergency landing in Buffalo, New York, after a passenger lost consciousness, but the individual received medical attention and was later released. The airline plans to transport the remaining passengers to their original destination in Hartford, Connecticut, on another aircraft. The Federal Aviation Administration initially reported a pressurization problem, but SkyWest and the FAA later stated that there was no indication of such an issue, and the FAA would investigate the incident.

**Our MoeSumm:** Passenger received medical attention before being released, airline says. SkyWest Airlines flight makes emergency landing in Buffalo, New York. The plane descended 28,000 feet in three minutes.

---

Figure 12: Case study on CNN/DM. **Errors** in each summary and the corresponding **related facts** are highlighted.

---

**Document:** jason terry , hero of dallas ' game one national basketball association finals win over miami , may need off-season surgery on an injured thumb that he has been nursing for four months .

**Ground Truth Summary:** mavs hero terry may need surgery to fix injured thumb

**Baseline BART-mix:** jason terry may need off-season surgery on an injured thumb that he has been nursing for four months. terry was hero of dallas' game one national basketball association finals win over miami.

**Baseline Prefix:** Jason terry may need off-season surgery on an injured thumb that he has been nursing for four months.

**GPT-3.5:** Jason Terry, the hero of Dallas' victory in game one of the National Basketball Association (NBA) finals against Miami, may require off-season surgery for a thumb injury he has been managing for four months.

**Our MoeSumm:** jason terry may need off-season surgery on an injured thumb.

---

Figure 13: Case study on Gigaword. Our MoeSumm generates a salient and concise summary.

**Document:** "I'm really looking forward to it - the home of Scottish football," said Rodgers ahead of his maiden visit. "I hear the pitch is good, a nice big pitch suits the speed in our team and our intensity." "The technical area goes right out to the end of the pitch, but you might need a taxi to get back to your staff." This will be Rodgers' second taste of the Old Firm derby and his experience of the fixture got off to a great start with a 5-1 league victory at Celtic Park last month. "It was a brilliant performance by the players in every aspect," he recalled. "Obviously this one is on a neutral ground, but we'll be looking to have a similar performance." "We'll be prepared and focused. We know it's going to be a tough game. We anticipated that the last time." Rodgers is also aware Celtic's visit to Hampden last season in the Scottish Cup semi-final against Rangers ended in defeat. "The last experience there wasn't so good for Celtic," he said. "But it's a different team with a different mentality and we'll look to take that mentality into the game." Rodgers lost two semi-finals in his time at Liverpool and is aiming to make it third time lucky at the club he joined in the summer. "You have to perform - that's what's important and if you get little bit of luck along the way then hopefully you'll get the result that you want." said the Northern Irishman. "So, for us, it's really looking at our performance level, which in the main has been at a really high level. My focus is always on my own team and, if we impose our style of play, it can give a lot of problems to opponents." Unlike predecessor Ronny Deila, Rodgers would not be drawn on this semi-final being a step on the way to a potential domestic treble. "It's very, very difficult to achieve," he said. "There's been great managers here in the past that haven't achieved that. But it's the first competition where we've a chance to win a trophy and I've always said that it's a priority for us because it's the first one. Our idea was to qualify for the Champions League, which we did, to make a good start in the league, which we've done, and then look to get to the League Cup final. Unless you can do that then you don't have the chance to win the trophies."

**Ground Truth Summary:** Manager Brendan Rodgers is sure Celtic can exploit the wide open spaces of Hampden when they meet Rangers in Sunday's League Cup semi-final.

**Baseline BART-mix:** Rangers host Celtic in the Scottish Cup semi-final at Hampden Park **on Saturday**.

**Baseline Prefix:** Celtic manager Brendan Rodgers has been tasked with leading his side to victory over Rangers in the Old Firm derby at Hampden Park.

**GPT-3.5:** Celtic manager Brendan Rodgers is excited about the upcoming Old Firm derby and believes the team's style of play will pose problems for opponents at the home of Scottish football, although he acknowledges the tough challenge it presents; he aims to draw on the team's improved mentality and past successes to secure a victory. While not openly discussing the potential of a domestic treble, Rodgers emphasizes the importance of performing well and focusing on one competition at a time, with winning the League Cup being a priority for Celtic this season.

**Our MoeSumm:** Celtic manager Brendan Rodgers is looking forward to playing in the Scottish Cup semi-final against Rangers.

Figure 14: Case study on XSum. **Errors** are highlighted.

