# OpenReview forum: "Enhancing Parameter Efficiency in Summarization via Expertise Separation"
_ICLR.cc/2024/Conference — Submitted to ICLR 2024_

### Official Review · Reviewer_wwvd · 2023-10-30

**Soundness:** 3 good
**Presentation:** 4 excellent
**Contribution:** 3 good
**Rating:** 6
**Confidence:** 3

**Summary:**

This work proposes a Mixture-of-Expert summarization architecture to address the summarization task. This approach utilizes a main expert for gaining the general summarization capability and deputy experts that selectively collaborate to meet specific summarization task requirements. Also, they propose a max-margin loss to distinguish the roles of the different experts. Experimental results show that this approach brings substantial improvements over strong baselines in both in-domain and out-of-domain scenarios across several widely-used benchmark datasets.

**Strengths:**

1. This paper is well-organized and easy to follow. Figures and tables are clearly presented. Sufficient references are discussed in related work.
2. The proposed approach is simple and easy to understand. Extensive experiments and analyses have been performed to confirm the effectiveness of this method. Specifically, this method performs well in both in-domain and out-of-domain scenarios.
3. Codes are provided in supplementary materials to ensure reproducibility.

**Weaknesses:**

1. **Limited Baselines**: some important baselines are omitted in the experiments. For example, [1] also used mixture-of-expert structure for abstractive summarization, but this work does not compare with it.
2. As this paper mentions the proposed method is **parameter-efficient**, experiments should include more than one base model to confirm this, otherwise we are not sure if this method only works on BART. For example, this paper can add PEGASUS as [1] used.

[1] Ravaut et al. Summareranker: A multi-task mixture-of-experts re-ranking framework for abstractive summarization. In Proc. of ACL, 2022.

**Questions:**

See weaknesses.

---

> ### Author Response · Authors · 2023-11-15
>
> Thank you for the valuable comments that help us improve the work. Below we address the concerns mentioned in the review:
>
> -----
>
> *Q1: Limited baselines*
>
> A1: we would like to mention that our compared baseline is not limited to BART. We also include prompt-tuning approach Prefix [1] and adapter-based baseline Lightweight [2], which all underperform our model significantly.
> For the Summareranker, we’d like to highlight the difference between these two models: Summareranker reranks the generated summaries from multiple models, while in our model, we have one unified model for multiple datasets and domains. Hence, we include it in our revision in related work.
>
>
> [1] Prefix-Tuning- Optimizing Continuous Prompts for Generation, ACL 2021
>
> [2] Lightweight Meta-Learning for Low-Resource Abstractive Summarization, SIGIR 2022
>
> ------
>
> *Q2: Using the method on other architecture such as PEGASUS*
>
> Thanks for your thoughtful consideration. In our Appendix A.4, we include the comparison with not only BART-large, but also BART-base and PEGASUS. The results show the generalization ability of our framework that does not depend on specific model scales or architectures. We choose the BART-large structure in our main experiment due to its better performance.
>
>
> -----
>
> Thank you again for the insightful questions. We hope that our answers and revised paper have addressed your questions.  If you have future suggestions or questions about our paper, we will feel very happy to share more responses.

---

> > ### Comment · Reviewer_wwvd · 2023-11-19
> >
> > Thanks for the rebuttal. I have read the responses from the authors as well as other reviews. I will keep my score.

---

> > > ### Author Response · Authors · 2023-11-20
> > >
> > > Thank you for your time and consideration of our rebuttal. Your feedback is invaluable and will guide our future improvements.

---

### Official Review · Reviewer_WEzF · 2023-10-31

**Soundness:** 3 good
**Presentation:** 3 good
**Contribution:** 2 fair
**Rating:** 6
**Confidence:** 3

**Summary:**

The paper addresses a narrow application of text summarization using the MoE method, in which the design is to have a 2-level hierarchy of application-specific deputy experts and  multi-mapped neural-network-style multi-purpose main experts. Using this design makes the method outperform other baselines including the canonical MoE in all datasets chosen, using traditional and adaptation evaluation settings. The experiments are conducted with 3 domain datasets. The authors also offer some insightful ablation studies concerning the design. Code is attached to the submission for reproducibility and the authors promise to release upon camera version.

**Strengths:**

- The paper has a good motivation with intuition design
- The paper is written well with strong and positive results
- Insightful analysis about the main components (architecture vs. max-margin loss) and other such as how deputies are separated/aligned to respective datasets
- Comprehensive and helpful appendices
- Provided code.

**Weaknesses:**

- Allocating each deputy to a single dataset doesn’t sound too practical and scalable as the number of datasets increases. One can question that if many datasets are similar, why are we not sharing similar deputies to many applications that are similar/related to each other?
- Although the authors offer explanations to Equations 8 and 9, it’s not totally clear about the design of the nonlinear function chosen in Equation 9. Why not offer ablation studies for choices of functions and constants involved?
- Also regarding the loss: I personally think one explanation for the max-margin loss is to battle with the imbalance, at the same time train the deputy to “spread-out” and learn all diverse data being fed. As a result, my hypothesis is with the current design, the imbalance loss from, say Gshard (and other methods could be used too in the RELATED WORK point below). Would be nice to study the ablations at some point.

- RELATED WORK: seems lack of many relevant work especially when it comes to imbalance issues:
  - Gshard
  - BASE layers: Simplifying training of large, sparse models.
  - Hash layers for large sparse models.
  - Mixture-of-experts with expert choice routing.
  - I suggest maybe should also see some updated related work including some from this year in new preprints such as in “Task-Based MoE for Multitask Multilingual Machine Translation”, in which the motivation is somewhat similar but the implementation seems more practical and scalable than the method in this paper (see weaknesses for more explanation on this).



- Human evaluation probably needs further clarification and only 3 of them do not seem convincing enough. Also, how were their employments made (see ethics section as well), and why only PhD students were chosen?
- BertScore is a decent choice, but I suggest since you already used GPT3.5, that you could use GPT3.5 or 4 to do the comparison as well, by having an instruction and ask the model to compare different models based on, e.g. 3 criteria in Table 2, then summarize the results. I think for now that would offer a very convincing evaluation.

**Questions:**

- In Table3, “MoESumm w/o DI”, how did you construct the max-margin loss without DI information?
- Please see more questions in Weaknesses.

**Details Of Ethics Concerns:**

The authors employ PhD students for human evaluation. Without further explanation on who they are and what are their relationships to the authors, I think it should be flagged for ethics review from AC.

---

> ### Author Response · Authors · 2023-11-15
>
> Thank you for the valuable comments that help us improve the work. Below we address the concerns mentioned in the review:
>
> -----
>
> *Q1: Allocating each deputy to a single dataset doesn’t sound too practical and scalable as the number of datasets increases. If many datasets are similar, why are we not sharing similar deputies?*
>
> We apologize for any confusion caused. Indeed, you are absolutely correct in noting that similar applications should share similar deputy experts, which aligns perfectly with our motivation for our MoeSumm structure. In practice, we do not allocate a specific deputy to a single dataset. Instead, we utilize shared multiple deputy experts which are dynamically selected based on the context and dataset category. We have revised and emphasized this aspect in Section 4 for clearer understanding.
>
> -----
> *Q2: The design of the nonlinear function in Equation 9.*
>
> We choose a Quintic function as it offers more stability according to previous study [1]. We now include a new ablation study in the revision:
>
> | Test | MoeSumm w/ Cube (R1/R2/RL/BS) | MoeSumm w/ Quintic (R1/R2/RL/BS) | MoeSumm w/ log (R1/R2/RL/BS) |
> |----|------|----|----|
> | CNN/DM       | 44.89/21.67/41.76/89.14       | 45.01/21.75/41.91/89.26        | 45.03/21.79/41.93/89.15     |
> | BIGPATENT    | 36.88/10.78/30.60/84.54       | 37.02/11.10/31.01/84.18        | 36.73/10.93/30.87/84.16     |
> | AESLC        | 32.73/17.99/30.86/86.15       | 32.87/17.96/30.92/86.54        | 32.66/18.33/30.29/86.21     |
>
> It can be seen that our MoeSumm keep relatively good performance on different nonlinear function design, particularly excelling with the Quintic function.
>
> [1] Prevent the Language Model from being Overconfident in Neural Machine Translation
>
> ------
> *Q3: Max-margin loss to solve the imbalance problem*
>
> A3: Thanks for listing the interesting references about the imbalance issue in MoE. However, we would like to clarify that our max-margin loss is not to balance between deputy experts. Instead, it is an adapted loss for hierarchical MoE structure, to differentiate between general ability in main expert and specialized ability in deputy experts. Hence, the listed works on imbalance loss can not replace our max-margin loss. Still, we are happy to include these references in our revision.
>
> ------
> *Q4:  Ablation study on the imbalance problem (max-margin loss)*
>
> A4: As explained in the previous answer, our max-margin loss is not designed for imbalance problem. In fact, we design a simple and intuitive way for the imbalance problem, which incorporates the dataset category information when selecting deputy experts.
> Our method is mostly related to [1. Task-Based MoE for Multitask Multilingual Machine Translation], where we both incorporate task(dataset) information into the selection process.
>
> Thanks for your suggestion, and we conduct new experiments to see if other imbalance-related techniques can be used to solve the imbalance problem in our scenario. Specifically, we choose the embedding method proposed in [1], which introduces task embedding, additional task gate, and MoE gate to selector the adaptors. The results show that the baseline and our methods achieve similar balanced distributions but with fewer parameters.
>
> | | Deputy Expert #1 | Deputy Expert #2 | Deputy Expert #3 |
> |---|---|---|---|
> | MoeSumm | 42% | 32% | 26%|
> | MoeSumm (adapter-version) | 40% | 35% | 24% |
>
> ----
> *Q5: Human evaluation details*
>
> We ensure high-quality annotations by hiring well-educated PhD students, specifically selecting those with backgrounds in computer science, chemistry, and biomedicine to guarantee diversity. Each case was compensated at \$ 0.01\$, leading to a total payment of \$6 per student.
>
> ----
> *Q6: Evaluation by LLM*
>
> Thanks to your advice, we conduct an evaluation with the latest GPT4. The instruction is:
> > Score summary1, summary2, and summary3 given the article in respect to fluency, with one (\*) to three stars (\***), where one star means bad and three stars means good. Only return the stars.
>
> |         | Succ | Inform | Flu  |
> |---|----|--|--|
> | BART-mix| 2.32 | 2.36   | 2.68 |
> | MoeSumm | 2.63 | 2.66   | 2.89 |
> | GPT-3.5 | 2.81 | 2.90   | 2.95 |
>
> The ranking aligns with our human assessment, affirming the reliability and effectiveness of both evaluation methods and showing our model's advantage.
>
> -----
> *Q7: MoESumm w/o DI details*
>
> By removing the dataset information in the selector, the dataset-aware selector degrades into a naive selector. That is, the $W_e$ matrix for $e$ dataset is changed to $W$, which is the same for all the datasets.
> Though the dataset information is removed from the selector, we still have the main and deputy experts, thus, the max-margin loss still works in differentiating these experts.
>
> -----
> We hope that our answers and revised paper have addressed your questions. If you have future suggestions or questions about our paper, we will feel very happy to share more responses.

---

> > ### Comment · Reviewer_WEzF · 2023-11-18
> >
> > Post rebuttal: I have read through other reviews and responses from the authors. I appreciate the efforts during the short period of time to make additional comments. Please revise the paper accordingly. In the meantime, I have increased my score to support the paper.
> >
> > I also like the niche area of summarization that the authors focus on. But w.r.t bias in human evaluation, I leave it for the AC to decide whether to accept the authors' clarification and official revision on that.

---

> > > ### Author Response · Authors · 2023-11-19
> > >
> > > We greatly appreciate that you have raised your score. Your feedback has been invaluable in refining our work.

---

### Official Review · Reviewer_gCW9 · 2023-11-01

**Soundness:** 3 good
**Presentation:** 4 excellent
**Contribution:** 3 good
**Rating:** 6
**Confidence:** 4

**Summary:**

This paper introduces MoeSumm, a mixture-of-experts approach to summarization that consists of a main expert model (arguably capturing general summarization skills) and "deputy experts", smaller domain-specific models. One key idea is to introduce max-margin loss term to jointly train the experts. Using diverse summarization benchmarks, the paper shows that MoeSumm outperforms a BART baseline in different in-domain and out-of-domain settings.

**Strengths:**

The paper addresses a relevant problem of parameter-efficient summarization using a novel method, and provide extensive experiments using automatic metrics and human evaluation showing the advantages of the approach. Ablation studies for key design decisions such as the max-margin loss term are also presented.

**Weaknesses:**

The proposed method presents some scaling challenges, which are partially discussed in Appendix A.7, stating that number of expert model parameters scale with the number of experts and datasets. Furthermore, there is a concatenation of parameters and activations in equations 6 and 7, which would result in more parameters. Consequently, when using experts, the concatenation of features makes MoeSumm a larger model than the BART-indiv baseline, which could explain the advantage of MoeSumm in the in-domain setting. Thus, it would be very informative to have a table with actual parameter count for each setting, especially during training (frequently the most important bottleneck).

Previous work that approaches parameter-efficient summarization by separating domain-specific concerns is not compared. FactorSum (Fonseca et al., 2022) also uses a BART backbone to capture general importance and uses a separate optimization procedure for domain-specific factors such as summary conciseness. It achieves superior performance (measured by ROUGE) than MoeSumm on arXiv and PubMed using less parameters (BART-base). In contrast, MoeSumm explores a wider range of domain-adaptation settings.

While I appreciate the analysis of dataset features in Section 5.2, item 3, more detailed analysis of the weight of each feature would be informative. Especially with respect to *summary length*, which are known to heavily affect ROUGE (Sun et al., 2019).

- Simeng Sun, Ori Shapira, Ido Dagan, and Ani Nenkova. 2019. How to Compare Summarizers without Target Length? Pitfalls, Solutions and Re-Examination of the Neural Summarization Literature.
- Fonseca, M., Ziser, Y., & Cohen, S. B. (2022). Factorizing content and budget decisions in abstractive summarization of long documents by sampling summary views.

**Questions:**

- Why do you think the zero-shot performance of MoeSumm is significantly better than BART-mix on some datasets (arXiv, MultiNews) but not others (Gigaword, Billsum)?

- Are differences of human evaluation scores (Table 2) statistically significant?

- What are the parameter counts for each model variant during training/inference? (see weakness comment above)

---

> ### Author Response · Authors · 2023-11-15
>
> Thank you for the valuable comments that help us improve the work. Below we address the concerns mentioned in the review:
>
> -----
> *Q1: it would be very informative to have a table with actual parameter count for each setting, especially during training (frequently the most important bottleneck).*
>
> A1: Thanks for your suggestion, below we show the parameter table and we also include it in the revision in Appendix 9:
>
>
> | Model Name            | BART-large | Prefix | MoeSumm (3 datasets) | MoeSumm (4 datasets) | MoeSumm (5 datasets) | MoeSumm (6 datasets) |
> |-----------------------|------------|--------|----------------------|----------------------|----------------------|----------------------|
> | Parameter Number      | 387.46M    | 449.42M (+16%) | 411.77M (+6%) | 411.84M (+6%) | 411.91M (+6%) | 411.98M (+6%) |
>
>
>
>
> It can be seen that further scaling up the training data does not significantly increase the number of parameters. We have included this table in the revised version.
>
> -----
> *Q2: FactorSum also uses a BART backbone to capture general importance and uses a separate optimization procedure for domain-specific factors such as summary conciseness. It achieves superior performance (measured by ROUGE) than MoeSumm on arXiv and PubMed using less parameters.*
>
> Thanks for bringing this great work to our attention, and we’ve included it in the revision! In the meantime, we would like to highlight the essential differences between our work:
>
> In FactorSum, the BART backbone model is trained on a single target dataset to acquire general summarization skills, where ‘general’ refers to summarization without specific consideration for their length. Consequently, a separate model is trained for each dataset to accommodate the different summarization requirements.
>
>
> In contrast, in our setting, the general summarization ability is not limited to one dataset and its length information, but it is learned from different datasets. Take Figure 5b in the paper for example, our specialized ability includes the understanding of the professional terms in a specific domain.
>
> -----
> *Q3: Analysis of the weight of each feature with respect to summary length*
>
> The summary length is indeed an important factor when evaluating the summary performance. As shown in Figure 5a, we can see that MoeSumm w/o deputy experts lacks information regarding the target length, while MoeSumm efficiently halts the generation process to produce an optimized summary length. The use of deputy experts is also linked to the length of summaries. For instance, our findings show that MoeSumm when paired with Deputy Expert #1 typically produces shorter sentences, averaging around 15 words, whereas MoeSumm working with Deputy Expert #3 tends to create longer sentences, averaging about 37 words.
>
> -----
> *Q4: Why the zero-shot performance of MoeSumm is significantly better than BART-mix on some datasets (arXiv, MultiNews) but not others (Gigaword, Billsum)?*
>
> Thank you for your insightful question. We believe that MoeSumm's effectiveness is attributed to reusing summarization skills from in-domain training datasets. For arXiv and MultiNews, which feature long, specialized content, skills learned from similar datasets like PubMed and CNN/DM are applicable. Conversely, Gigaword and Billsum present unique challenges due to their brevity and legislative content, not covered in our training. Following this direction, we draw the performance on Gigaword as training datasets get larger. As shown in Appendix A.5, MoeSumm's performance on Gigaword notably improves (by 7% in ROUGE-1 scores) when trained with similar datasets such as AESLC and Reddit.
>
> ------
>
> *Q5: Are differences in human evaluation scores (Table 2) statistically significant?*
>
> Yes, the improvements to the second-best performance are statistically significant for $\alpha=0.01$.
>
> ------
> Thank you again for the insightful questions. We hope that our answers and revised paper have addressed your questions.  If you have future suggestions or questions about our paper, we will feel very happy to share more responses.

---

> > ### Comment · Reviewer_gCW9 · 2023-11-18
> >
> > Thank you the the clarifications, especially regarding the model scaling with respect to the number of experts. I will update my scores accordingly.

---

> > > ### Author Response · Authors · 2023-11-19
> > >
> > > Thanks for your support of our work. We believe that your valuable comments have improved the paper.

---

### Official Review · Reviewer_BPBZ · 2023-11-05

**Soundness:** 1 poor
**Presentation:** 3 good
**Contribution:** 3 good
**Rating:** 5
**Confidence:** 4

**Summary:**

In the quest for a summarization model that balances flexibility and adaptability without the extensive parameter scaling characteristic of large language models (LLMs), this paper introduces MoeSumm, a Mixture-of-Expert Summarization architecture. The model is premised on the notion that while a core summarization competence is universally applicable across tasks, domain-specific nuances require tailored expertise. MoeSumm is structured with a central 'main expert' responsible for general summarization, complemented by 'deputy experts' that are invoked for task-specific challenges. To enhance the model’s capability for domain differentiation, a max-margin loss is employed, encouraging a clearer separation between general and specialized summarization skills.

**Strengths:**

The idea of using a core network (main expert) for the generalizability of the summarization and multiple experts for different domain is clear and reasonable. The proposed max-margin loss also ensures that the model does not overly rely on the main expert. Extensive experimental results show that the proposed method outperforms the baselines in three settings, i.e., in-domain, out-of-domain, and zero-shot.

**Weaknesses:**

1. The major weakness is that the baselines used in this paper is relatively weak, i.e., BART (Lewis et al., 2020). It is suggested to compare with the state-of-the-art approaches, e.g., [A,B], to show this paper actually improves the state-of-the-art models. For example, in [B], the performance on Reddit100 is 34.24 in terms of R1, whereas the result in this paper is 25.57. Although this result may relate to the training strategy, it is suggested to align the setting for a fair comparison.
2. The efficiency is not the superiority of the paper since many paper with few-shot setting directly tune the models. MoE models are usually good in the performance but worse in the efficiency, similar with the ensemble methods.
3. The contribution is relatively minor. It is suggested to try some alternative designs for justifying the proposed approach.

[A] T. Yu, Z. Liu, and P. Fung. AdaptSum: Towards Low-Resource Domain Adaptation for Abstractive Summarization. In Proceedings of the Conference of the North American Chapter of the Association for Computational Linguistics: Human Language Technologies, pages 5892–5904, 2021.

[B] Y. -S. Chen, Y. -Z. Song and H. -H. Shuai, "SPEC: Summary Preference Decomposition for Low-Resource Abstractive Summarization," in IEEE/ACM Transactions on Audio, Speech, and Language Processing, vol. 31, pp. 603-618, 2023.

**Questions:**

The titles in the paper and system are different, i.e., "Enhancing Parameter Efficiency in Summarization via Expertise Separation" and "Flexible and Adaptable Summarization via Expertise Separation". It is suggested to select one and revise the paper accordingly.

---

> ### Author Response · Authors · 2023-11-15
>
> Thank you for the valuable comments that help us improve the work. Below we address the concerns mentioned in the review:
>
> -----
> *Q1: The compared BART baseline is weak. Should include [Adaptsum] and [SPEC].*
>
> A1: Thank you very much for bringing these papers to our attention. First, we would like to mention that our compared baseline is not limited to BART. We also include strong baselines including prompt-tuning approach Prefix [1] and adapter-based baseline Lightweight [2], which all underperform our model significantly. This is introduced in the paper and details can be found in Appendix.
> For [Adaptsum], it utilizes more training datasets (300 vs 10) and its performance is lower than our model and baseline Lightweight in different datasets such as on Reddit and AESLC, as shown below. For [SPEC], it proposes several criteria for choosing meta training data, while in our work, we do not specifically choose the training data but use a unified one. Our main target is not to improve performance by data selection or data augmentation. Instead, we aim to separate general and specific summarization abilities. Hence, the result is not directly comparable. Following your suggestion, we now include the two works and the discussion in our revision.
>
> | Model/RG-1 score        | Reddit | AESLC |
> |--------------|--------|-------|
> | AdaptSum     | 23.25  | 26.97 |
> | Lightweight  | 24.47  | 32.02 |
> | MoeSumm      | 25.57  | 32.87 |
>
>
> [1] Prefix-Tuning- Optimizing Continuous Prompts for Generation, ACL 2021
>
> [2] Lightweight Meta-Learning for Low-Resource Abstractive Summarization, SIGIR 2022
>
> -----
> *Q2: MoE models are worse in efficiency in comparison with existing few-shot baselines that directly finetune the model.*
>
> A2: Firstly, we want to clarify that our model has fewer parameters compared with a number of existing few-shot baselines. As shown in the table, our model has 10% less parameters compared with the recent baseline Prefix and 24% less parameters with Lightweight.
>
> Secondly, our model is different from existing MoE models. Our MoeSumm is a hierarchical structure with main and deputy experts. The general summarization ability for various domains is shared to achieve parameter efficiency.
>
> | Model Name            | Parameter Number         |
> |-----------------------|--------------------------|
> | BART-large            | 387.46M                  |
> | Prefix                | 449.42M (+16%)           |
> | Lightweight           | 508.62M (+30%)           |
> | MoeSumm  | 411.77M (+6%)            |
>
>
> ------
>
> *Q3: Contributions of the paper*
>
> A3: The contribution of the paper can be summarized into three aspects:
> (1) We propose MoeSumm, a parameter-efficient summarization model that is  applicable to a variety of in-domain summarization tasks and is well-suited for out-of-domain few-shot and zero-shot summarization scenarios. The model's parameter efficiency is ensured by the shared general summarization ability. (2) To achieve the above goal, we design MoeSumm with a mixture-of-expert structure and a max-margin loss to distinguish the roles of the different experts. (3) Experimental results demonstrate that   MoeSumm brings substantial improvements over strong baselines in both in-domain and out-of-domain scenarios across benchmark datasets.
>
> -----
> *Q4: The title in the system and the title in the paper are not consistent.*
>
> A4: Thank you for the reminder. We’ll keep the title in the paper ‘Flexible and Adaptable Summarization via Expertise Separation’.
>
> -----
> Thank you again for the insightful questions. We hope that our answers and revised paper have addressed your questions.  If you have future suggestions or questions about our paper, we will feel very happy to share more responses.

---

> > ### Comment · Reviewer_BPBZ · 2023-11-19
> >
> > In fact, Table 1 shows the main results, which only contain BART-mix, BART-indiv, and FlatMoe (simple MOE design). It is still suggested to include SOTA approaches (some papers in 2023 would be great as the topic (few-shot/zero-shot summary) is not new). On the other hand, reducing 10% parameters may not be significant. After reading the rebuttal and comments from other reviewers, some of my concerns are addressed. I would like to raise the rating to 5. Thanks for the clarification.

---

> > > ### Author Response · Authors · 2023-11-20
> > >
> > > Thank you so much for the response. Here, we address your concerns in detail below.
> > >
> > > *W1: Compare with papers in 2023, e.g., "SPEC: Summary Preference Decomposition for Low-Resource Abstractive Summarization," in IEEE/ACM Transactions on Audio, Speech, and Language Processing, 2023.*
> > >
> > > A1: Thanks for the listed work. We are trying to reproduce the model in this reference and compare our work with it. However, we encountered multiple challenges as the authors have not released the model checkpoint or the meta data used for different target datasets. We have reached out to the authors to request these files and are actively pursuing this to ensure a thorough and fair comparison.
> > >
> > > In the meantime, we want to highlight that our compared baselines are competitive and representative of current advancements in the field. For example, the Lightweight baseline outperforms [UNISUMM: Unified Few-shot Summarization with Multi-Task Pre-Training and Prefix-Tuning, ACL 2023] significantly on different datasets. Specifically, it achieves 35.54/13.94/27.79 on XSum and 25.37/7.05/19.81 on Reddit, outperforming UNISUMM's scores of 33.33/11.36/25.85 and 24.54/6.17/18.30, respectively.
> > >
> > >
> > > ------
> > >
> > > *W2: Reducing 10% parameters may not be significant*
> > >
> > > A2: Thank you for your feedback. We wish to highlight the parameter efficiency of our model compared with a number of baselines. It saves 10% more parameters than [Prefix, ACL 2021], 24% more than [Lightweight, SIGIR 2022], and 34% more than [Efficient framework for low-resource abstractive summarization by meta-transfer learning and pointer-generator networks, 2023]. This significant reduction in parameters, while maintaining performance, marks a noteworthy achievement in summarization research.
> > >
> > > Moreover, we would like to highlight that our key contribution lies in the novel design of a hierarchical mixture of expert structure with expertise separation. This design enables MoeSumm to utilize a main expert for general summarization capabilities while employing deputy experts for specific summarization tasks. This hierarchical approach distinguishes our work from existing MoE methods in the field.
> > >
> > > ----
> > >
> > > We sincerely hope our responses have addressed your concerns.

---

### Meta-Review · Area_Chair_bBZN · 2023-12-13

**Metareview:**

All reviewers acknowledged the motivation behind utilizing a Mixture of Experts (MoE) architecture and associated max-margin loss for summarization domain adaptation.  I disregarded comments that make little or no sense (e.g., comparing with Summareranker or SPEC). However, considering its marginal gain, not considering LLMs (new ones already use an MoE), and the datasets used for experiments (XSum and Wikihow), I am not 100% sure about its potential impact in the field.

**Justification For Why Not Higher Score:**

NA

**Justification For Why Not Lower Score:**

NA

---

### Decision · Program_Chairs · 2024-01-16

Reject